# AniTalker: Animate Vivid and Diverse Talking Faces through Identity-Decoupled Facial Motion Encoding

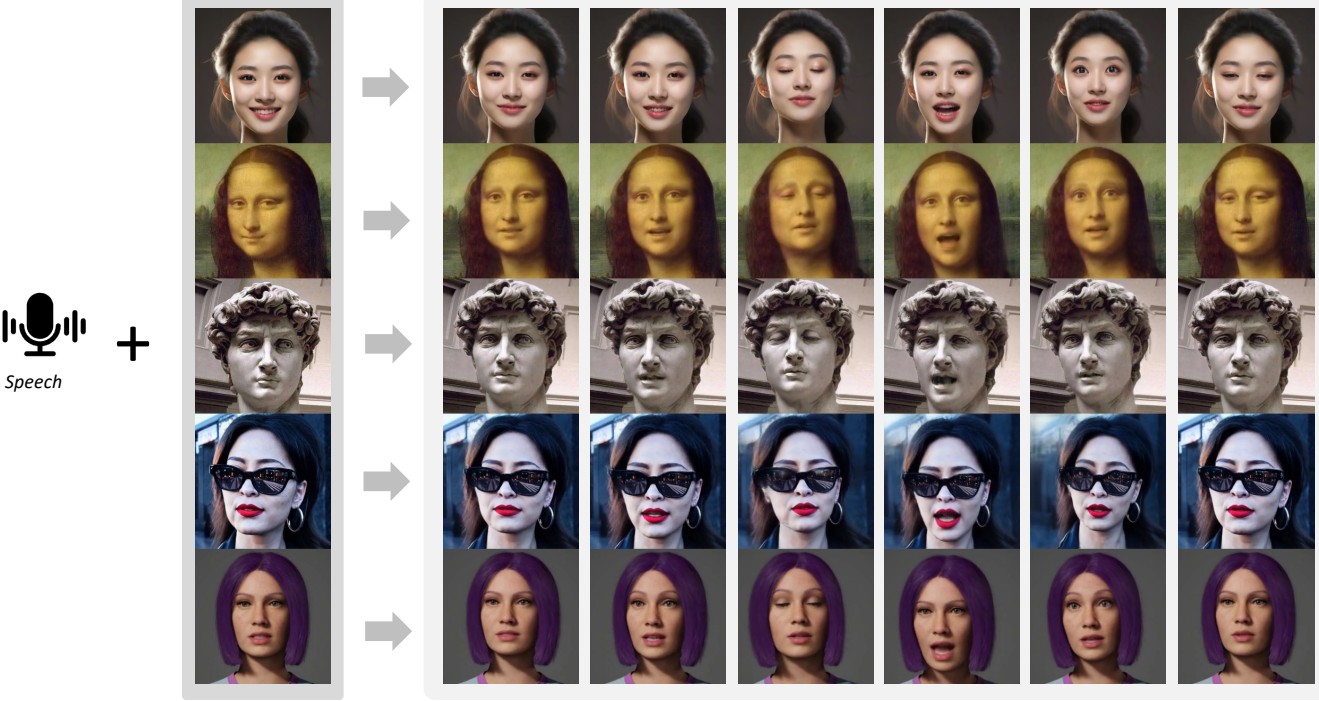

**Figure 1: We introduce AniTalker, a framework that transforms a single static portrait and input audio into animated talking videos with naturally flowing movements. Each column of generated results utilizes identical control signals with similar poses and expressions but incorporates some random variations, demonstrating the diversity of our generated outcomes.**

## ABSTRACT

The paper introduces AniTalker, an innovative framework designed to generate lifelike talking faces from a single portrait. Unlike existing models that primarily focus on verbal cues such as lip synchronization and fail to capture the complex dynamics of facial expressions and nonverbal cues, AniTalker employs a universal motion representation. This innovative representation effectively captures a wide range of facial dynamics, including subtle expressions and head movements. AniTalker enhances motion depiction through two self-supervised learning strategies: the first involves reconstructing target video frames from source frames within the same identity to learn subtle motion representations, and the second develops an identity encoder using metric learning while actively minimizing mutual information between the identity and motion encoders. This approach ensures that the motion representation is dynamic and devoid of identity-specific details, significantly reducing the need for labeled data. Additionally, the integration of a diffusion model with a variance adapter allows for the generation of diverse and controllable facial animations. This method not only demonstrates AniTalker's capability to create detailed and realistic facial movements but also underscores its potential in crafting dynamic avatars for real-world applications. Synthetic results can be viewed at https://anitalker.github.io.

## CCS CONCEPTS

• **Computing methodologies** → **Motion capture**; **Procedural animation**.

## KEYWORDS

Talking Face, Self-supervised, Motion Encoding, Disentanglement

*ACM MM, 2024, Melbourne, Australia*
© 2024 Copyright held by the owner/author(s). Publication rights licensed to ACM.
ACM ISBN 978-x-xxxx-xxxx-x/YY/MM
https://doi.org/10.1145/nnnnnnn.nnnnnnn

**Unpublished working draft. Not for distribution.**

# 1 INTRODUCTION

Integrating speech signals with single portraits [13, 18, 31, 42, 44, 56–58] to generate talking avatars has greatly enhanced both the entertainment and education sectors, providing innovative avenues for interactive digital experiences. While current methodologies [34, 44, 54, 58, 59] have made notable strides in achieving synchronicity between speech signals and lip movements, thus enhancing verbal communication, they often neglect the critical aspect of nonverbal communication. Nonverbal communication encompasses the transmission of information without the use of words, including but not limited to specific head movements, facial expressions, and blinking. Research [33] indicates that these nonverbal cues are pivotal in communicating.

The primary challenge lies in the inadequacy of existing models to encapsulate the complex dynamics associated with facial motion representation. Existing approaches predominantly employ explicit structural representations such as blendshapes [3, 13, 32], landmark coefficients [18, 45, 57], or 3D Morphable Models (3DMM) [7, 14, 25] to animate faces. Designed initially for single-image processing, these methods offer a constrained approximation of facial dynamics, failing to capture the full breadth of human expressiveness. Recent advancements [11, 24] have introduced trainable facial motion encoders as alternatives to conventional explicit features, showing significant progress in capturing detailed facial movements. However, their deployment is often tailored for specific speakers [11] or limited to the mouth region [24], highlighting a gap in fine-grained motion representation that captures all varieties of facial dynamics.

A universal and fine-grained motion representation that is applicable across different characters remains absent. Such a representation should fulfill three key criteria: capturing minute details, such as minor mouth movements, eye blinks, or slight facial muscle twitching; ensuring universality, making it applicable to any speaker while removing identity-specific information to maintain a clear separation between appearance and motion; and incorporating a wide range of nonverbal cues, such as expressions, head movements, and posture.

In this paper, we introduce **AniTalker**. Our approach hinges on a universal motion encoder designed to grasp the intricacies of facial dynamics. By adopting the self-supervised learning paradigm, we mitigate the reliance on labeled data, enabling our motion encoder to learn robust motion representations. This learning process operates on dual levels: one entails understanding motion dynamics through the transformation of a source image into a target image, capturing a spectrum of facial movements, from subtle changes to significant alterations. Concurrently, the use of identity labels within the dataset facilitates the joint optimization of an identity recognition network in a self-supervised manner, further aiming to disentangle identity from motion information through mutual information minimization. This ensures that the motion representation retains minimal identity information, upholding its universal applicability.

To authenticate the versatility of our motion space, we integrate a diffusion model and a variance adapter to enable varied generation and manipulation of facial animations. Thanks to our sophisticated representation and the diffusion motion generator, AniTalker is capable of producing diverse and controllable talking faces.

In summary, our contributions are threefold:

(1) We have developed universal facial motion encoders using a self-supervised approach that effectively captures facial dynamics across various individuals. These encoders feature an identity decoupling mechanism to minimize identity information in the motion data and prevent identity leakage.

(2) Our framework includes a motion generation system that combines a diffusion-based motion generator with a variance adapter. This system allows for the production of diverse and controllable facial animations, showcasing the flexibility of our motion space.

(3) Extensive evaluations affirm our framework's contribution to enhancing the realism and dynamism of digital human representations, while simultaneously preserving identity.

# 2 RELATED WORKS

**Speech-driven Talking Face Generation** refers to creating talking faces driven by speech, We categorize the models based on whether they are single-stage or two-stage. Single-stage models [34, 55, 58] generate images directly from speech, performing end-to-end rendering. Due to the size constraints of rendering networks, this method struggles with processing longer videos, generally managing hundreds of milliseconds. The two-stage type [3, 11, 13, 18, 24, 31, 57] decouples motion information from facial appearance and consists of a speech-to-motion generator followed by a motion-to-video rendering stage. As the first stage solely generates motion information and does not involve the texture information of the frames, it requires less model size and can handle long sequences, up to several seconds or even minutes. This two-stage method is known to reduce jitter [3, 11, 24], enhance speech-to-motion synchronization [11, 13, 31, 57], reduce the need for aligned audio-visual training data [3, 24], and enable the creation of longer videos [18]. Our framework also employs a two-stage structure but with a redesigned motion representation and generation process.

**Motion Representation** serves as an essential bridge between the driving features and the final rendered output in creating talking faces. Current methods predominantly utilize explicit structural representations, such as blendshapes [3, 13, 30], 3D Morphable Models (3DMMs) [25], or landmarks [45, 57]. These formats offer high interpretability and facilitate the separation of facial actions from textures, making them favored as intermediary representations in facial generation tasks. However, due to the wide range of variability in real-world facial movements, they often fail to capture the subtle nuances of facial expressions fully, thus limiting the diversity and expressiveness of methods dependent on these representations. Our research is dedicated to expanding the spectrum of motion representation by developing a learned implicit representation that is not constrained by the limitations of explicit parametric models.

**Self-supervised motion transfer approaches** [29, 38, 41, 45, 46, 48, 51] aim to reconstruct the target image from a source image by learning robust motion representations from a large amount of unlabeled data. This significantly reduces the need for labeled data. A key challenge in these methods is separating motion from identity information. They primarily warp the source image using predicted dense optical flow fields. This approach attempts to disentangle motion from identity by predicting distortions and

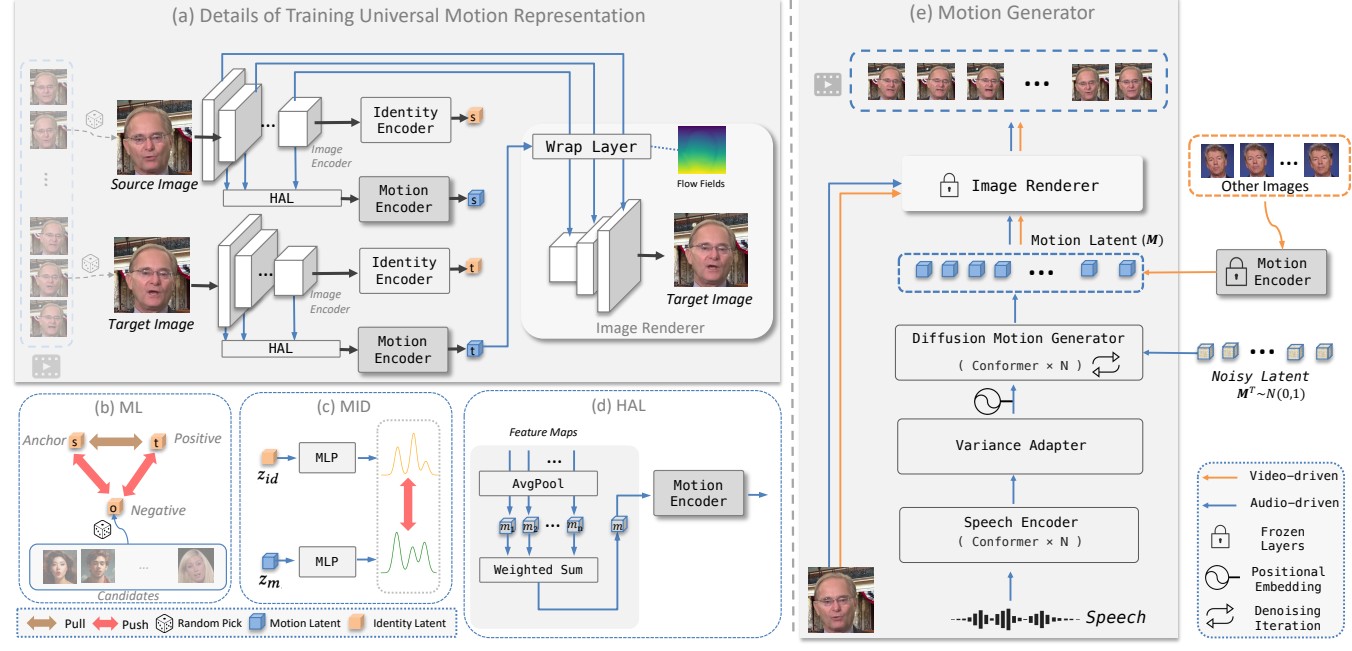

**Figure 2: The AniTalker framework comprises two main components: learning a universal motion representation and then generating and manipulating this representation through a sequence model. Specifically, the first part aims to learn a robust motion representation by employing metric learning (ML), mutual information disentanglement (MID), and Hierarchical Aggregation Layer (HAL). Subsequently, this motion representation can be used for further generation and manipulation.**

transformations of the source image. However, information leakage occurs in practice, causing the target image to contain not just motion but also identity information. Building on this observation, we explicitly introduce identity modeling and employ the Mutual Information Neural Estimation (MINE) [1, 4] method to achieve a motion representation independent of identity.

**Diffusion Models** [19] have demonstrated outstanding performance across various generative tasks [12, 17, 21, 36]. Recent research has utilized diffusion models as a rendering module [2, 11, 24, 27, 37, 40, 42]. Although diffusion models often produce higher-quality images, they require extensive model parameters and substantial training data to converge. To enhance the generation process, several approaches [18, 25, 26, 30, 52] employ diffusion models for generating motion representations. Diffusion models excel at addressing the one-to-many mapping challenge, which is crucial for speech-driven generation tasks. Given that the same audio clip can lead to different actions (e.g., lip movements and head poses) across different individuals or even within the same person, diffusion models provide a robust solution for managing this variability. Additionally, the training and inference phases of diffusion models, which systematically introduce and then remove noise, allow for the incorporation of noise during generation to foster diversity. We also use diffusion in conjunction with our motion representation to further explore diversity in talking face generation.

## 3 ANITALKER FRAMEWORK

### 3.1 Model Overview

AniTalker contains two critical components: (1) Training a motion representation that can capture universal face dynamics, and (2) Based on the well-trained motion encoder from the previous step, the generation or manipulation of the motion representation using the user-controlled driving signal to produce the synthesised talking face video.

### 3.2 Universal Motion Representation

Our approach utilizes a self-supervised image animation framework, employing two RGB images from a video clip: a source image $I^s$ and a target image $I^t$ ($I \in \mathbb{R}^{H \times W \times 3}$), to serve distinct functions: $I^s$ provides identity information, whereas $I^t$ delivers motion details. The primary aim is to reconstruct $I^t$. Due to the random selection of frames, occasionally adjacent frames are chosen, enabling the network to learn representations of subtle movements. As depicted in Figure 2 (a), both the source and target images originate from the same video clip. Through this self-supervised learning method, the target image's encoder is intended to exclusively capture motion information. By learning from frame-to-frame transfer, we can acquire a more universal representation of facial motion. This representation includes verbal actions such as lip movements, as well as nonverbal actions, including expressions, posture, and movement.

To explicitly decouple motion and identity in the aforementioned processes, we strengthen the self-supervised learning approach

by incorporating Metric Learning (ML) and Mutual Information Disentanglement (MID). Specifically:

**Metric Learning.** Drawing inspiration from face recognition [8, 43] and speaker identification [9], metric learning facilitates the generation of robust identity information. This technique employs a strategy involving pairs of positive and negative samples, aiming to minimize the distance between similar samples and maximize it between dissimilar ones, thereby enhancing the network's ability to discriminate between different identities. This process can also proceed in a self-supervised fashion, with each iteration randomly selecting distinct identities from the dataset. Specifically, the approach establishes an anchor ($a$) and selects a positive sample ($p$) and a negative sample ($n$)—corresponding to faces of different identities—with the goal of reducing the distance ($d$) between the anchor and the positive sample while increasing the distance between the anchor and the negative samples. This optimization, depicted in Figure 2 (b), involves randomly selecting a different identity from a list of candidates not belonging to the current person as the negative sample. The optimization goal for this process is as follows:

$$\mathcal{L}_{ML} = \max\left(0,\, d(a, p) - d(a, n) + \text{margin}\right)$$

Here, the margin is a positive threshold introduced to further separate the positive and negative samples, thus improving the model's ability to distinguish between different identities.

**Mutual Information Disentanglement.** Although metric learning effectively constrains the identity encoder, focusing solely on this encoder does not adequately minimize the identity information within the motion encoder. To tackle this issue, we utilize Mutual Information (MI), a statistical measure that evaluates the dependency between the outputs of the identity and motion encoders. Given the challenge of directly computing MI between two variables, we adopt a parametric method to approximate MI estimation among random variables. Specifically, we use CLUB [4], which estimates an upper bound for MI. Assuming the output of the identity encoder is the identity latent $z_{id}$ and the motion encoder's output is the motion latent $z_m$, our goal is to optimize the mutual information $I(\mathcal{E}(z_{id}); \mathcal{E}(z_m))$, where $\mathcal{E}$ denotes the learnable Multi-Layer Perceptron (MLP) within CLUB. This optimization ensures that the motion encoder primarily captures motion, thereby preventing identity information from contaminating the motion space. This strategy is depicted in Figure 2 (c).

In summary, by leveraging Metric Learning and Mutual Information Disentanglement, we enhance the model's capacity to accurately differentiate between identity and motion while reducing reliance on labeled data.

**Hierarchical Aggregation Layer (HAL).** To enhance the motion encoder's capability to understand motion variance across different scales, we introduce the Hierarchical Aggregation Layer (HAL). This layer aims to integrate information from various stages of the image encoder, each providing different receptive fields [23]. HAL processes inputs from all intermediate layers of the image encoder and passes them through an Average Pooling (AvgPool) layer to capture scale-specific information. A Weighted Sum [50] layer follows, assigning learnable weights to effectively merge information from these diverse layers. This soft fusion approach enables the motion encoder to capture and depict movements across a broad range of scales. Such a strategy allows our representations to adapt to faces of different sizes without the need for prior face alignment or normalization.

Specifically, the features following the AvgPool layer are denoted as $[m_1, m_2, \ldots, m_n]$, representing the set of averaged features, with $[w_1, w_2, \ldots, w_n]$ as the corresponding set of weights, where $n$ symbolizes the number of intermediate layers in the image encoder. These weights undergo normalization through the softmax function to guarantee a cumulative weight of 1. The equation for the weighted sum of tensors, indicating the layer's output, is formulated as $\mathbf{m} = \sum_{i=1}^{n} w_i \cdot m_i$. The softmax normalization process is mathematically articulated as $w_i = \frac{e^{W_i}}{\sum_{j=1}^{n} e^{W_j}}$, ensuring the proportional distribution of weights across the various layers. Subsequently, $\mathbf{m}$ is fed into the motion encoder for further encoding.

**Learning Objective.** The main goal of learning is to reconstruct the target image by inputting two images: the source and the target within the current identity index. Several loss functions are utilized during the training process, including reconstruction loss $L_{recon}$, perceptual loss $L_{percep}$, adversarial loss $L_{adv}$, mutual information loss $L_{MI}$, and identity metric learning loss $L_{ML}$. The total loss is formulated as follows:

$$L_{motion} = L_{recon} + \lambda_1 L_{percep} + \lambda_2 L_{adv} + \lambda_3 L_{MI} + \lambda_4 L_{ML}$$

## 3.3 Motion Generation

Once the motion encoder and image renderer are trained, at the second stage, we can freeze these models. The motion encoder is used to generate images, then video-driven or speech-driven methods are employed to produce motion, and finally, the image renderer carries out the final frame-by-frame rendering.

### 3.3.1 Video-Driven Pipeline.
Video driving, also referred to face reenactment, leverages a driven speaker's video sequence $\mathbf{I}^d = [I_1^d, I_2^d, \ldots, I_T^d]$ to animate a source image $I^s$, resulting in a video that accurately replicates the driven poses and facial expressions. In this process, the video sequence $\mathbf{I}^d$ is input into the motion encoder, previously trained in the first phase, to extract the motion latent. This latent, along with $I^s$, is then directly fed, frame by frame, into the image renderer for rendering. No additional training is required. The detailed inference process, where the orange lines represent the data flow during video-driven inference, is depicted in Figure 2 (e).

### 3.3.2 Speech-Driven Pipeline.
Unlike video-driven methods that use images, the speech-driven approach generates videos consistent with the speech signal or other control signals to animate a source image $I^s$. Specifically, we utilize a combination of diffusion and variance adapters: the former learns a better distribution of motion data, while the latter mainly introduces attribute manipulation.

**Diffusion Models.** For generating motion latent sequences, we utilize a multi-layer Conformer [16]. During training, we incorporate the training process of diffusion, which includes both adding noise and denoising steps. The noising process gradually converts clean Motion Latent $\mathbf{M}$ into Gaussian noise $\mathbf{M}^T$, where $T$ represents the number of total denoising steps in the diffusion process. Conversely, the denoising process systematically eliminates noise from the Gaussian noise, resulting in clean Motion Latents. This iterative process better captures the distribution of motion, enhancing

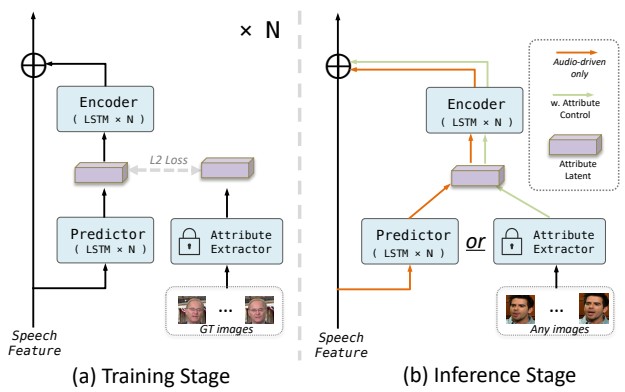

**Figure 3: Variance Adapter Block. Each block models a single attribute and can be iterated multiple times, where $N$ represents the number of attributes.**

the diversity of the generated results. During the training phase, we adhere to the methodology described in [19] for the DDPM's training stage, applying the specified simplified loss objective, as illustrated in Equation 1, where $t$ represents a specific time step and $\mathbf{C}$ represents the control signal, which refers to either speech or speech perturbed by a Variance Adapter (to be discussed in the following section). For inference, considering the numerous iteration steps required by diffusion, we select the Denoising Diffusion Implicit Model (DDIM) [39]—an alternate non-Markovian noising process—as the solver to quicken the sampling process.

$$L_{\text{diff}} = \mathbb{E}_{t,\mathbf{M},\epsilon} \left[ \| \epsilon - \hat{\epsilon}_t \left( \mathbf{M}_t, t, \mathbf{C} \right) \|^2 \right] \tag{1}$$

**Variance Adapter.** The Variance Adapter [35] is a residual branch connected to audio features, allowing optional control over the speech signal. Originally proposed to mitigate the one-to-many problem in Text-to-Speech (TTS) tasks, its architecture includes a predictor and an encoder that use speech signals to predict attribute representations. A residual connection is then applied between the encoder output and the speech signals. During the Training Stage, the encoder processes speech features in collaboration with the predictor to minimize the L2 loss against a ground truth control signal. This includes incorporating an attribute extractor for targeting specific attributes, such as employing a pose extractor (yaw, pitch, roll) to control head posture during the audio generation process. In the Inference Stage, the trained encoder and predictor can flexibly synthesize speech with controlled attributes or operate based on speech-driven inputs. The detailed structure is depicted in Figure 3. Our approach extends previous works [11, 18] by incorporating LSTM [15] for improved temporal modeling and introducing additional cues such as head position and head scale, which we refer to as camera parameters. The architecture is detailed in Figure 3.

**Learning Objective.** The total loss comprises diffusion loss and variance adapter loss, where $K$ represents the number of attributes:

$$L_{\text{gen}} = L_{\text{diff}} + \lambda \sum_{k=1}^{K} L_{\text{var}_k}$$

# 4 EXPERIMENTS

## 4.1 Experimental Settings

We utilizes three datasets: VoxCeleb [28], HDTF [56], and VFHQ [49]. Due to different processing approaches across these datasets, we re-downloaded the original videos and processed them in a unified way. Specifically, our processing pipeline included filtering out blurred faces and faces at extreme angles. It is noted that we did not align faces but instead used a fixed detection box for each video clip, allowing for natural head movement. This effort resulted in a dataset containing 4,242 unique speaker IDs, encompassing 17,108 video clips with a cumulative duration of 55 hours. Details of this filtering process are provided in the supplementary material. Each video in these datasets carries a unique facial ID tag, which we used as labels for training our identity encoder. We also reserved some videos from HDTF for testing, following the test split in [55].

**Scenario Setting** We evaluate methods under two scenarios: video-driven and speech-driven, both operating on a one-shot basis with only a single portrait required. The primary distinction lies in the source of animation: image sequences for video-driven and audio signals for speech-driven scenarios. The detailed data flow for inference is illustrated in Figure 2. Additionally, each scenario is divided into two types: self-driven, where the source and target share the same identity, and cross-driven, involving different identities. In speech-driven tasks, if posture information is needed, it is provided from the ground truth. Moreover, for our motion generator, unless specified otherwise, we use a consistent seed to generate all outcomes. To ensure a fair comparison, the output resolution for all algorithms is standardized to $256 \times 256$.

**Implementation Details** In training the motion representation, our self-supervised training paradigm is primarily based on LIA [46]. Both the identity and motion encoders employ MLPs. Our training targets use the CLUB [1] for mutual information loss, in conjunction with AAM-Softmax [43]. This robust metric learning method utilizes angular distance and incorporates an increased number of negative samples to enhance the metric learning loss. In the second phase, the speech encoder and the Motion Generator utilize a four-layer and a two-layer conformer architecture, respectively, inspired by [11, 24]. This architecture integrates the conformer structure [16] and relative positional encoding [6]. A pre-trained HuBERT-large model [20] serves as the audio feature encoder, incorporating a downsampling layer to adjust the audio sampling rate from 50 Hz to 25 Hz to synchronize with the video frame rate. The training of the audio generation process spans 125 frames (5 seconds). Detailed implementation specifics and model structure are further elaborated in the supplementary materials.

**Evaluation Metric** For **objective metrics**, we utilize Peak Signal-to-Noise Ratio (PSNR), Structural Similarity Index (SSIM) [47], and Learned Perceptual Image Patch Similarity (LPIPS) [53] to quantify the similarity between generated and ground truth images. Cosine Similarity (CSIM) [2] measures facial similarity using a pretrained face recognition. Lip-sync Error Distance (LSE-D) [5] assesses the alignment between generated lip movements and the corresponding audio. Regarding **subjective metrics**, we employ the Mean Opinion Score (MOS) as our metric, with 10 participants

---

[1] https://github.com/Linear95/CLUB/
[2] https://github.com/dc3ea9f/vico_challenge_baseline

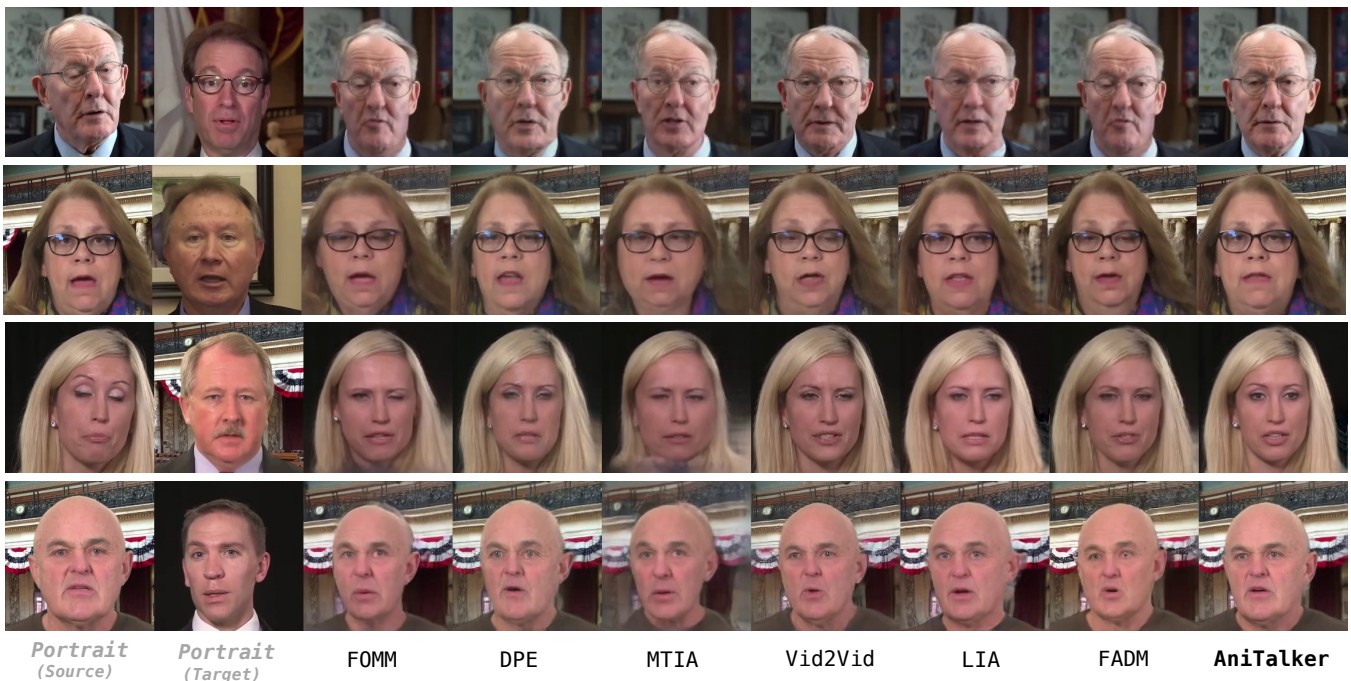

Portrait *(Source)*    Portrait *(Target)*    FOMM    DPE    MTIA    Vid2Vid    LIA    FADM    **AniTalker**

**Figure 4: Cross-Reenactment Visualization: This task involves transferring actions from a target portrait to a source portrait to evaluate each algorithm's ability to separate motion and appearance. Starting from the third column, each column represents the output from a different algorithm. The results highlight our method's superior ability to preserve fidelity in both motion transfer and appearance retention.**

rating our method based on Fidelity (F), Lip-sync (LS), Naturalness (N), and Motion Jittering (MJ).

## 4.2 Video Driven Methods

**Table 1: Quantitative comparisons with previous Face Reenactment methods.**

| Method | Self-Reenactment | | | | Cross-Reenactment | | |
|---|---|---|---|---|---|---|---|
| | PSNR↑ | SSIM↑ | LPIPS↓ | CSIM↑ | SSIM↑ | LPIPS↓ | CSIM↑ |
| FOMM [38] | 23.944 | 0.775 | 0.178 | 0.830 | 0.411 | 0.423 | 0.494 |
| DPE [29] | 27.239 | 0.861 | 0.151 | 0.912 | 0.445 | 0.410 | 0.567 |
| MTIA [41] | 28.435 | 0.870 | 0.122 | **0.929** | 0.393 | 0.456 | 0.448 |
| Vid2Vid [45] | 27.659 | 0.870 | 0.115 | 0.924 | 0.410 | 0.401 | 0.553 |
| LIA [46] | 25.854 | 0.831 | 0.137 | 0.916 | 0.421 | 0.406 | 0.522 |
| FADM [51] | 26.169 | 0.849 | 0.147 | 0.916 | 0.445 | 0.399 | 0.574 |
| **AniTalker** | **29.071** | **0.905** | **0.079** | 0.927 | **0.494** | **0.347** | **0.586** |

**Quantitative Results** We benchmarked our approach against several leading face reenactment methods [29, 38, 41, 45, 46, 51], all employing variations of self-supervised learning. The results are presented in Table 1. Due to the inherent challenges and the absence of frame-by-frame ground truth in Cross-Reenactment (using another person's video for driving), the overall results tend to be lower compared to Self-Reenactment (using the current person's video). In Self-Reenactment, our algorithm achieved superior results for image structural metrics such as PSNR, SSIM, and LPIPS, validating

the effectiveness of our motion representation in reconstructing images. Additionally, using the CSIM metric to measure face similarity, we observed that the similarity between the reconstructed face and the original portrait was the second highest, slightly behind MTIA [41], illustrating our model's identity preservation capabilities. For Cross-Reenactment, where the portrait serves as ground truth and considering cross-driven deformations, we focused on high-level metrics: SSIM and LPIPS. Our method demonstrated commendable performance. We also evaluated CSIM, which, unlike self-reenactment, showed a significant improvement, achieving the best results among these datasets. This highlights our algorithm's outstanding ability to disentangle identity and motion when driving with different individuals.

**Qualitative Results** To highlight comparative results, we conducted a cross-reenactment scenario analysis with different algorithms, as presented in Figure 4. The objective was to deform the source portrait using the actions of the target. Each row in the figure represents a driving case. We observed that baseline methods exhibited varying degrees of identity leakage, where the identity information from the target contaminated the source portrait's identity. For example, as demonstrated in the fourth row, the slim facial structure of the driving portrait led to slimmer outcomes, which was unintended. However, our results consistently preserved the facial identity. Additionally, in terms of expression recovery, as evident in the first and third rows, our approach replicated the action of opening the eyes in the source portrait accurately, creating a natural set of eyes. In contrast, other algorithms either produced

slight eye-opening or unnatural eyes. These qualitative findings highlight the advantage of decoupling ability.

## 4.3 Speech-driven Methods

**Table 2: Quantitative comparisons with previous speech-driven methods. The subjective evaluation is the mean option score (MOS) rated at five grades (1-5) in terms of Fidelity (F), Lip-Sync (LS), Naturalness (N), and Motion Jittering (MJ).**

| Method | Subjective Evaluation | | | | Objective Evaluation (Self) | | |
|---|---|---|---|---|---|---|---|
| | MOS-F↑ | MOS-LS↑ | MOS-N↑ | MOS-MJ↑ | SSIM↑ | CSIM↑ | Sync-D↓ |
| MakeItTalk [59] | 3.434 | 1.922 | 2.823 | 3.129 | 0.580 | 0.719 | 8.933 |
| PC-AVS [58] | 3.322 | 3.785 | 2.582 | 2.573 | 0.305 | 0.703 | **7.597** |
| Audio2Head [44] | 3.127 | 3.650 | 2.891 | 2.467 | 0.597 | 0.719 | 8.197 |
| SadTalker [54] | 3.772 | 3.963 | 2.733 | 3.883 | 0.504 | 0.723 | 7.967 |
| **AniTalker** | **3.832** | **3.978** | **3.832** | **3.976** | **0.671** | **0.725** | 8.298 |

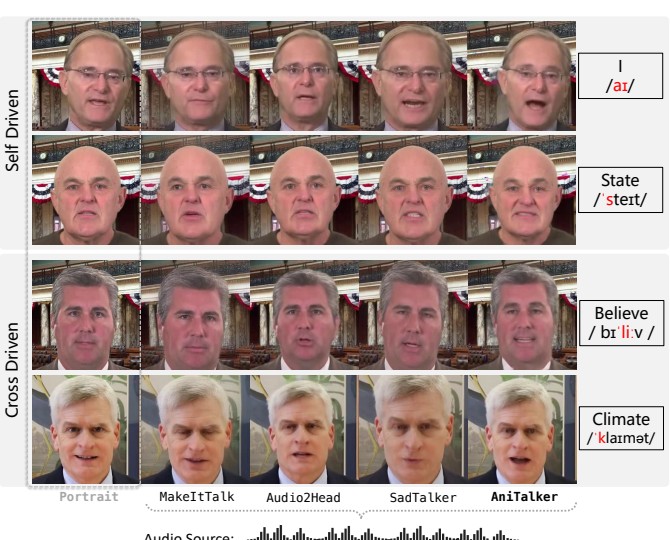

**Figure 5: Visual comparison of the speech-driven method in self- and cross-driven scenarios. Phonetic sounds are highlighted in red.**

We compare our method against existing state-of-the-art speech-driven approaches, including MakeItTalk [59], PC-AVS [58], Audio2Head [44], and SadTalker [54]. **Quantitative results** are presented in Table 2. From the subjective evaluation, our method consistently shows improvements in fidelity, lip-sync accuracy, naturalness, and a reduction in motion jittering, particularly noted for the enhanced naturalness of movements. These advancements can be attributed to our sophisticated universal motion representation. The objective evaluation involves driving the image with its audio. Compared to these methods, our approach shows significant improvements in SSIM and CSIM. However, our Sync-D metric shows a decrease, which we believe is due to two main reasons: (1) we do not use this metric as a supervisory signal, and (2) the Sync-D metric focuses on short-term alignment and does not adequately

represent long-term information that is more crucial for the comprehensibility of generated videos. This is also corroborated by the **qualitative results** shown in Figure 5, highlighting our model's ability to produce convincingly synchronized lip movements to the given phonetic sounds.

## 4.4 Ablation Study

**Table 3: Quantitative comparisons of disentanglement methods and the HAL module in Self-Reenactment setting**

| Method | ML | MID | HAL | PNSR ↑ | SSIM ↑ | CSIM ↑ |
|---|---|---|---|---|---|---|
| Baseline | | | | 25.854 | 0.849 | 0.916 |
| Triplet [10] | ✓ | | | 26.455 | 0.860 | 0.911 |
| AAM-Softmax [43] | ✓ | | | 27.922 | 0.894 | 0.923 |
| AAM-Softmax + CLUB [4] | ✓ | ✓ | | 28.728 | 0.900 | 0.924 |
| **AniTalker** | ✓ | ✓ | ✓ | **29.071** | **0.905** | **0.927** |

*4.4.1 Ablations on Disentanglement.* To further validate the effectiveness of our disentanglement between motion and identity, we conducted tests using various methods. Initially, to evaluate the performance of developing a reliable identity encoder using only Metric Learning (ML) without Mutual Information Disentanglement (MID), we assessed both Triplet loss [10] and AAM-Softmax [43]. Our results indicate that AAM-Softmax, an angle-based metric, achieves superior outcomes in our experiments. Additionally, by incorporating a mutual information decoupling module alongside AAM-Softmax, we noted further improvements in results. This enhancement encouraged the motion encoder to focus exclusively on motion-related information. These findings are comprehensively detailed in Table 3.

**Table 4: Different intermediate representations under the Self-Reenactment setting. 'Face Repr.' is short for face representation, and 'Dim.' represents the corresponding dimension.**

| Method | Face Repr. | Dim. | PSNR ↑ | SSIM ↑ | CSIM↑ |
|---|---|---|---|---|---|
| EMOCA [7] | 3DMM | 50 | 20.911 | 0.670 | 0.768 |
| PIPNet [22] | Landmark | 136 | 22.360 | 0.725 | 0.830 |
| **AniTalker** | Motion Latent | 20 | **29.071** | **0.905** | **0.927** |

*4.4.2 Ablation Study on Motion Representation.* To compare our motion representation with commonly used landmark and 3D Morphable Model (3DMM) representations, we utilized 68 2D coordinates [22] (136 dimensions) for the landmark representation and expression parameters (50 dimensions) from EMOCA [7] for the 3DMM representation. In self-reenactment scenarios, all rendering methods were kept consistent, and different features were used to generate driven images. We observed several key points: (1) As shown in Table 4, our learned representation exhibits a more compact dimensionality, indicating a more succinct encoding of facial dynamics. (2) Our video comparisons show that, unlike these explicit representations, our implicit motion representation maintains

frame stability without the need for additional smoothing. This can be attributed to our self-supervised training strategy of sampling adjacent frames, which effectively captures subtle dynamic changes while inherently ensuring temporal stability.

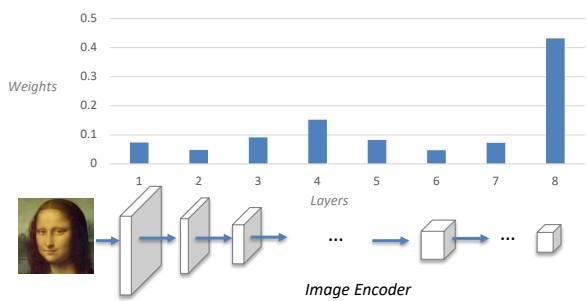

**Figure 6: The weights of motion representation from different layers of the Image Encoder.**

*4.4.3 Ablations on HAL.* To explore the significance of the Hierarchical Aggregation Layer (HAL) in dynamic representations, we conducted a series of ablation experiments focusing on the HAL layer. The results showed that models incorporating the HAL layer exhibited performance improvements, as detailed in the final row of Table 3. To analyze the impact and importance of different HAL layers on motion representation, we extracted and examined the softmax-normalized weights of each layer (a total of 8 layers in our experiment) in our Image Encoder as shown in Figure 6. It was found that the weights of the last layer contributed most significantly, likely because it represents global features that can effectively recover most motion information. Notably, the fourth layer—situated in the middle of the image encoder feature map—demonstrated a local maximum. Considering the receptive field size of this layer's patch is similar to the size of eyes and approximately half the size of the mouth, this finding suggests that the layer plays a potential role in simulating areas such as the mouth and eyes. These results not only confirm the pivotal role of the HAL layer in dynamic representation but also reveal the deep mechanisms of the model's ability to capture facial movements of different scales.

## 5 DISCUSSION

**Discussion on Universal Motion Representation** Our investigations into the model's ability to encode facial dynamics have highlighted a universal representation of human facial movements. As depicted in Figure 7, we observed that different individuals maintain consistent postures and expressions (such as turning the head left, speaking with homophones, and closing eyes) at each point within our motion space, demonstrating that our motion space forms a Motion Manifold. This manifold facilitates the representation of a continuous motion space, enabling the precise modeling of subtle facial feature variations and allowing for smooth transitions. Additionally, by integrating perturbations through diffusion noise, our model can simulate random, minute motion changes that align with fundamental movement patterns, thus enhancing the diversity

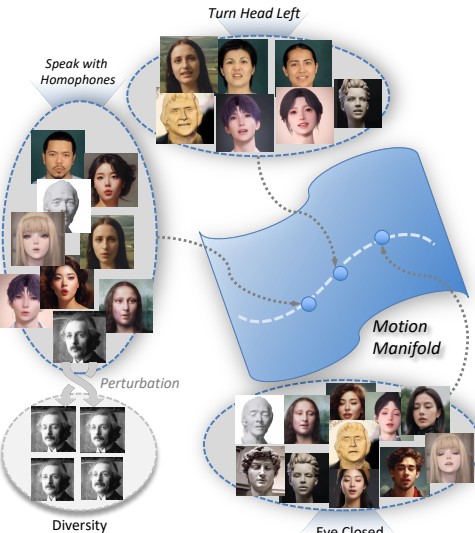

**Figure 7: Motion Manifold of the continuous motion space.**

of generated expressions. These findings demonstrate that our motion representation has a robust capacity to capture and represent a wide array of human facial movements.

**Discussion on Generalization Ability** Although our model is trained on real human faces, it demonstrates the ability to generalize to other images with facial structures, such as cartoons, sculptures, reliefs, and game characters. This underscores the model's excellent scalability. We primarily attribute this capability to the complete decoupling of identity and motion, which ensures that the model grasps the intrinsic nature of facial movements, thereby enhancing its generalization capability.

## 6 CONCLUSION

The AniTalker framework represents a significant advancement in the creation of lifelike talking avatars, addressing the need for a fine-grained and universal motion representation in digital human animation. By integrating a self-supervised universal motion encoder and employing sophisticated techniques like metric learning and mutual information disentanglement, AniTalker effectively captures the subtleties of both verbal and non-verbal facial dynamics. The resulting framework not only achieves enhanced realism in facial animations but also demonstrates strong generalization capabilities across different identities and media. AniTalker sets a new benchmark for the realistic and dynamic representation of digital human faces, promising broad applications in entertainment, communication, and education.

**Limitation and Future Work** While AniTalker shows promise in generalizing motion dynamics, it still faces challenges. Our rendering network generates frames individually, which can lead to inconsistencies in complex backgrounds. Additionally, limited by the performance of the warping technique, extreme cases where the face shifts to a large angle may result in noticeable blurring at the edges. Future work will focus on improving the temporal coherence and rendering effects of the rendering module.

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
