# OpenReview forum: "AniTalker: Animate Vivid and Diverse Talking Faces through Identity-Decoupled Facial Motion Encoding"
_acmmm.org/ACMMM/2024/Conference — MM2024 Poster_

### Official Review · Reviewer_Yf2S · 2024-04-28

**Rating:** 4
**Confidence:** 4

**Summary:**

The paper introduces AniTalker, a framework for generating lifelike talking faces from a single portrait. Unlike existing models that focus primarily on lip synchronization and overlook the complexity of facial expressions and nonverbal cues, The proposed work employs a universal motion representation to capture a wide range of facial dynamics, including subtle expressions and head movements.

AniTalker enhances motion depiction through two self-supervised learning strategies: first, by employing metric learning (ML) to encode identity-related information from the source face image frame, and second, by developing a motion encoder using Mutual Information Disentanglement to separate spatial information within the same identity. To ensure that motion representation remains dynamic and free of identity-specific details while reducing reliance on labeled data.

Additionally, the work integrates a diffusion model with a variance adapter, enabling the generation of diverse and controllable facial animations from the audio as a reference source.

**Strengths:**

1. The related work is well-discussed, providing a comprehensive overview of existing models and their limitations in capturing facial dynamics.
2. The rendered results exhibit high fidelity and realism, indicating the effectiveness of the proposed work in generating lifelike facial animations.
3. The encoder modules utilized for extracting identity and spatial information are well-described, with additional support from the Hierarchical Aggregation Layer (HAL), enhancing the understanding of the underlying mechanisms driving motion depiction

**Limitations:**

1. The second pipeline utilized for rendering frames from the audio sequence lacks proper description and explanation. There is insufficient detail provided on how the training of the diffusion module is conducted, how denoising is performed to render the sequence of frames.
2. The presentation of the second pipeline in Figure 2 (e), representing the framework of the model, is unclear and incomplete. It fails to provide a comprehensive overview of the second pipeline's architecture, leading to confusion regarding its purpose and necessity.
3. The necessity of introducing the second pipeline for rendering frames from the audio sequence is not adequately justified. There is a lack of clarity on how this pipeline enhances the overall performance or capabilities of the model, raising questions about its relevance and contribution to the proposed framework.
4. The absence of detailed explanations and clear presentation of the second pipeline contributes to the complexity and confusion surrounding its implementation and effectiveness. This can hinder the understanding of the model's operation and impede its usability and reproducibility.
5. To address these weaknesses, the authors should provide comprehensive descriptions of the training process for the diffusion-based module, elucidate the denoising methods utilized.  Additionally, enhancing the presentation of the second pipeline in visual representations such as figures or diagrams can improve its clarity and facilitate better understanding for readers.

**Suitability:**

2

---

### Official Review · Reviewer_dkjp · 2024-05-05

**Rating:** 5
**Confidence:** 2

**Summary:**

This paper presents an innovative method for talking face generation. Given a target face image and a motion control signal (video or speech), the method enables high-quality talking face generation. By learning a universal motion representation, the method can capture diverse facial motions.

**Strengths:**

1. The supplementary videos are excellent, which will contribute to the research on talking face generation.
2. The paper is written in a clear and logical manner. Sufficient experimental results are shown in the paper to demonstrate the validity of the proposed method.
3. A training strategy that effectively decouples identity information from motion information is proposed in the paper. By utilising metric learning and mutual information decoupling, the strategy can effectively suppress the leakage of identity information into motion information.

**Limitations:**

1. The generated result is $256 \times 256$ resolution, which can be extended to $512 \times 512$ or even higher resolution in the future research.
2. For some examples of complex backgrounds, the generated results still show background shaking, which can be further optimized.
3. In the introduction to metric learning in Section 3.2 (lines 360-364), the method requires the estimation of an anchor, but it is not stated that the anchor is an image with the same identity as the positive sample? In addition, there are similarities between the method and triplet loss, please elaborate on the differences if possible.

**Suitability:**

3

---

### Official Review · Reviewer_1mt1 · 2024-05-24

**Rating:** 3
**Confidence:** 4

**Summary:**

This paper proposes a new speech driven talking face generation method AniTalker which employs a universal motion representation to capture a wide range of facial dynamics, including subtle expressions and head movements.

**Strengths:**

- The new universal facial motion encoders are a good way to disentangle identity and motion from input portrait images, especially the introduction of MI and MID modules.
- Paper writing is good to follow and the content is well organized. Paper contains many details about training, data cleaning, module design, etc.

**Limitations:**

- What are the flow fields, how are they computed and what about their functions in the image renderer module?
- In the image renderer, looks like it inputs only the target image's motion latent from the motion encoder, without the source image's motion latent? If so, how to measure the motion difference between source and target image?
- Does this paper use the pretrained CLUB model to compute MI between identity code and motion code? If so, can the CLUB method handle any general type and any dimensional data? Why is this measurement reliable on the identity code and motion code originally encoded from images?
- Figure 6 and Section 4.4.3 analyzes the impact of HAL layers. From them, I can only find the different impacts from different layers, but why does it prove HAL is helpful for dynamic representation? And why can the normalized weights' magnitude show the impact of the layer?
- One suggestion is to add some descriptions about reconstruction loss, perceptual loss, adversarial loss used in training, since they are also important parts of the final loss. I can see from supplemental material that they just follow the definition of LIA, but still it worth noticing how they are constructed.
- How are head poses introduced into the network? Does the network use original groundtruth head poses or predict poses from input audio/video directly? There are some descriptions from Section B2 in supplemental material, but still no information about how to use head poses in the main pipeline. Meanwhile, from supplemental videos I can see the results have the same head poses for different images input. Are these poses some groundtruth poses or predicted?
- Experiments are quite thorough but miss some recent speech-driven methods to compare. Most of the compared method in this paper are a little old. Here I added some new ones as below. And actually DiffTalk [37] is also a recent method, but authors only cite it without detailed comparisons.
- Supplemental videos don't contain comparisons with previous methods. Impossible to see if AniTalker surpasses others or not.

References:
- [CVPR 2023] PD-FGC: Progressive Disentangled Representation Learning for Fine-Grained Controllable Talking Head Synthesis
- [AAAI 2023] StyleTalk: One-shot Talking Head Generation with Controllable Speaking Styles
- [ICCV 2023] Efficient Emotional Adaptation for Audio-Driven Talking-Head Generation
- [CVPR 2023] TalkLip: Seeing What You Said: Talking Face Generation Guided by a Lip Reading Expert

**Suitability:**

2

---

### Meta-Review · Area_Chair_q5sJ · 2024-07-01

**Recommendation:** Accept (Poster)
**Confidence:** 5

**Metareview:**

AniTalker proposes a novel framework for generating talking faces from a single portrait and audio input. The reviewers acknowledge the clarity of writing, promising results (dkjp, Yf2S), effectiveness of decoupled motion representation (1mt1, dkjp), and well-discussed related work (Yf2S). However, there are concerns regarding the completeness of explanations, particularly for the second rendering pipeline, and the lack of comparison with recent methods (1mt1, Yf2S).

**Strengths:**

* Clear and well-organized writing (dkjp, Yf2S).
* High-quality talking face generation (dkjp, Yf2S).
* Effective decoupling of identity and motion information (1mt1, dkjp).
* Comprehensive overview of related work (Yf2S).

**Weaknesses:**

* **Second Rendering Pipeline:**
    * Lack of explanation for training and denoising (Yf2S).
    * Unclear justification for necessity (Yf2S).
    * Unclear presentation in Figure 2(e) (Yf2S).
* **Comparisons:** Limited comparison with recent speech-driven methods (1mt1).

While AniTalker presents a promising approach, the need for a clearer explanation of the second rendering pipeline and a more comprehensive comparison with recent work prevent its acceptance for oral presentation. However, the overall quality of the research and the reviewers' generally positive evaluations make it suitable for poster presentation. This will allow the authors to showcase their work and receive feedback from the conference attendees.

The authors should consider revising their paper to address the following before the conference:

* Provide a comprehensive description of the training process for the diffusion-based module, including the denoising methods used.
* Enhance the clarity of Figure 2(e) to better represent the second pipeline's architecture.
* Include comparisons with recent speech-driven talking face generation methods in the final version of the paper (supplemental material is acceptable).